# Alkali-Activated Stainless Steel Slag as a Cementitious Material in the Manufacture of Self-Compacting Concrete

**DOI:** 10.3390/ma14143945

**Published:** 2021-07-14

**Authors:** Julia Rosales, Francisco Agrela, José Luis Díaz-López, Manuel Cabrera

**Affiliations:** Construction Engineering Area, Rabanales Campus, Leonardo da Vinci Building, University of Córdoba, 14071 Córdoba, Spain; jrosales@uco.es (J.R.); ep2diloj@uco.es (J.L.D.-L.); manuel.cabrera@uco.es (M.C.)

**Keywords:** stainless steel slag, alternative alkaline activation, self-compacting concrete, mechanical behaviour, durability properties

## Abstract

This work develops the manufacture of self-compacting concrete (SCC) with 50% cement reduction. As an alternative binder to cement, the viability of using an alkali-activated combination of stainless steel slag (SSS) and fly ash (FA) has been demonstrated. SSS was processed applying three different treatments. Binders were manufactured mixing 35% SSS with 65% FA, as precursors, and a hydroxide activating solution. This binder was replaced by the 50% cement for the manufacture of SCC. The results obtained show good mechanical properties and durability. The study shows a reduction in the use of cement in the manufacture of SCC reusing two wastes.

## 1. Introduction

Advances in research to find new materials with better characteristics, more economical processes, and less pollution have been developing over many years. Within these lines of research, we are facing the challenge of finding substitute materials for the most used construction material, ordinary Portland cement (OPC). Geopolymers are expected to be the cementitious material of the future. Their characteristics mean that they can be converted into a partial, or total, substitute for cement.

Geopolymers form a subset within the typology of alkaline-activated cementitious materials. These materials show different properties and applications, being an alternative to OPC, resulting in less use of natural resources, energy, and CO_2_ emissions [1].

In recent years, geopolymers have been developed as a material to stabilise solid, hazardous, and nuclear waste [2,3,4,5,6,7].

In addition, geopolymers have cementitious properties comparable to those of OPC. They are seen as an alternative to OPC due to the lower emission (up to 80% less) of CO_2_ in their manufacture [8,9]. Furthermore, they show improvements in the mechanical properties and durability of the manufactured material [10,11] such as good resistance to freeze-thaw cycles and to the action of acids; low permeability [12]; and easy adhesion in glasses, ceramic materials, concretes, and steels.

For the improvement of knowledge and the development of new raw materials in the preparation of alkali-activated binders, as a substitute for OPC, several works are studying different by-products as new precursors and activators [13]. Among these alternatives, some industrial waste products, called supplementary cementitious materials (SCMs), can be used to partially, or completely, substitute OPC in concrete production [14].

For the correct alkaline activation of a material, it has been demonstrated that the presence of silicates is necessary. In recent years, researchers have developed studies for the replacement of synthetic alkaline silicates with high-silica-content waste.

Among the main waste considered to be a source of silica are ashes from agro-industrial waste, e.g., rice husk ashes, sugar cane straw ashes, or biomass ashes. All these ashes have been studied by different researchers, and it has been observed that they contain important chemical elements from the point of view of cementing materials such as silicon, aluminium, and sodium/potassium [15,16,17,18,19,20].

Although many studies have focused on the use of ashes from different industrial activities, among the different waste materials, several of the metallurgical activities have also been studied, mainly due to their amorphous morphology that gives them reactive hydraulic properties. Compressive strength results for geopolymers made from metallurgical slag showed positive results.

Metallurgical wastes such as SCMs include the following: zinc slag presenting iron, calcium, aluminium, silicon, and zinc in the form of oxides. With the application of zinc slag in geopolymer binder, mixes with a compressive strength of more than 50 MPa at 7 days and more than 70 MPa at 180 days were achieved [21].

Another type of metallurgical slag analysed by previous authors was nickel and magnesium slag, with a majority composition of SiO_2_ (52.3%) and MgO (29.6%), this slag showed a mainly amorphous structure and combined with coal fly ash, resulted in 60 MPa compressive strength in geopolymers manufactured [22]. Another type of waste studied from the metallurgical sector was smelting slag, characterised by a low CaO content and a main crystalline phase of iron silicates. This morphological characteristic showed a decrease in compressive strength [23,24]. Among the studies of alkaline activation of metallurgical wastes, the most common are those of ladle furnace slag and electric arc furnace slag. The characteristics of these slags are a high percentage of calcium (CaO, 54.5%) and also the presence of SiO_2_ (16.4%), Al_2_O_3_ (11.1%), and Fe_2_O_3_ (8.7%). Using these, good compressive strength results were obtained in the geopolymers [25,26,27,28].

Studies of the possibility of using activated steel slag in the manufacture of concrete are scarce. Most studies focus on the application of this waste as a substitute for aggregates in the manufacture of concrete [29,30].

The results obtained in relation to the activation of metallurgical waste led us to analyse the waste produced from one of the most dynamic production sectors, stainless steel slag (SSS).

As a novelty in this work, activated stainless steel slag was used not as a substitute for aggregates in the manufacture of self-compacting concrete (SCC) but as a substitute for cement and limestone filler.

The chemical properties of SSS show an activation potential, in addition to having shown (in previous studies) their viability for use as a cement substitute, resulting in a better mechanical behaviour [31]. In this study, we evaluated the alkaline activation of SSS in combination with fly ash (FA) for partial OPC substitution in concrete manufacture. Different pretreatments, different percentages of SSS, and different curing conditions were studied to evaluate the mechanical behaviour of the binder. Subsequently, a novel study was carried out in which the mixtures with the best results were used as a substitute for OPC and filler in the manufacture of SCC. This resulted in an alkali-activated SCC with a reduction of up to 50% of cement.

## 2. Experimental Programme

This study evaluated SSS with different treatments to assess its potential for alkaline activation and use in the manufacture of SCC as OPC and filler substitute.

Two phases of study were carried out. The first phase was to evaluate the possibility of alkaline activation of SSS in the manufacture of binders, selecting the optimal substitution ratios and treatments based on the results of mechanical behaviour obtained and then the application of the selected mixtures in the manufacture of SCC (Figure 1).

Geopolymer pastes with different dosages of SSS and FA were manufactured to achieve the optimum ratio, and the pastes were evaluated under different curing conditions. NaOH 8 molar was used as an activating solution. The evaluation of the possible activation was carried out through a study of the compressive and flexural strength of the manufactured pastes.

Following the evaluation of the geopolymer binder, a study was carried out on the incorporation of binders with optimum dosage of SSS-FA, as a cement substitute and filler, in the manufacture of SCC. The mechanical behaviour of the alkali-activated SCC was analysed.

### 2.1. Materials

In the present study, four different types of SSS processes were used; non-processed (SSS-NP), crushed and sieved SSS evaluated in order to obtain the fraction 0/125 µm (SSS-C), SSS burned at a temperature of 800 °C for 18 h (SSS-B), and SSS-CB (stainless steel slag with combined crushing and burning treatment). The stainless steel slag (SSS) came from the company Acerinox Europa S.A.U., located in Los Barrios (Cádiz), Spain.

The chemical and physical properties of the different SSS samples are summarised in Table 1. Additionally, the properties of the FA used are shown.

SSS-NP was characterised by its high contents of CaO and SiO_2_: 49.09% and 28.3%, respectively. The compositions of SSS-C, SSS-B, and SSS-CB were similar, although there were slight variations in the CaO, SiO_2_, and MgO contents. The proportions of these elements were higher with respect to SSS-NP.

Furthermore, the SSS treatment decreased the density and water absorption.

Regarding the properties of FA, all samples of SSS showed higher values of density and absorption, as well as an increase in the values of CaO and MgO. On the contrary, the values of SiO_2_ and Al_2_O_3_ were lower in the SSS in relation to that in FA.

The high oxide content in SSS, such as CaO, SiO_2_, and MgO, show a chemical potential that can be activated and exploited [34].

A mineralogical study of the SSS was carried out using powder X-ray diffraction analysis with Bruker equipment (Karlsruhe, Germany) (model D8 Advance A25) and a Cu twist tube (XRD,). SSS showed a mostly crystalline morphology (Figure 2), this aspect being a disadvantage for their activation [35,36].

The composition of SSS was mainly merwinite (Ca_3_Mg(SiO_4_)_2_), calcium-magnesium-silica oxide, and akermanite (Ca–Mg–Si). This mineralogy corresponds to the elemental components shown in Table 1.

The natural pozzolans used for the manufacture of cements are characterised by a high SiO_2_ content [37]. A high SiO_2_ content of natural pozzolan implies a high strength of pozzolanic activity in cements [38].

Traditionally, the pozzolanic activity of materials was established according to certain limits in their chemical composition [39]. The chemical composition of FA and SSS in relation to the ranges established for pozzolanic materials is shown in Figure 3.

It can be observed that FA is within the limits established for SiO_2_ and CaO; however, SSS with all treatments applied was shown to be outside these limits, having lower values of SiO_2_ and higher values of CaO than traditional pozzolans. Recent studies have shown that ash with high calcium content show pozzolanic and hydraulic behaviour [40]. Therefore, the high values of CaO obtained from SSS would make possible its alkaline activation, resulting in a material with cementing activity.

To verify this pozzolanic activity, a pozzolanicity study was carried out on each of the materials in accordance with the UNE-EN 196-5:2011 standard.

Figure 4 shows the results of the Frattini test at 8 and 14 days plotted on the [OH^−^] vs. [CaO] diagram.

SSS with different treatments applied showed negative pozzolanity at 8 and 14 days. However, the application of a crushing treatment (SSS-C) showed positive pozzolanity after 14 days.

According to Frattini’s test, it can be seen from Figure 1 that the data obtained on the variation of [OH^−^] and [CaO] show that the pozzolanic activity of SSS-C is high, while SSS-NP, SSS-B, SSS-CB, and FA are less reactive. SSS-C showed low [OH^−^] and [CaO] values that place this slag in the “pozzolanic region” established by Frattini (Figure 4). Higher values of [OH^−^] and [CaO] concentrations in the rest of the slags analysed place them in the “non-pozzolanic region” showing a lower activation potential.

According to previous studies [39], SSS follow a pattern of behaviour according to pathway B established by Pourkhorshidi et al. [41] in which it is established that this is the pathway followed by highly active pozzolans. In the pozzolanicity evaluation diagram (Figure 4), SSS and FA show a reaction of similar trajectory, in which the values move to lower alkalinity and lower Ca ion concentration as time progresses.

This behaviour is due to pozzolans such as silica fume or fly ash.

### 2.2. Mix Proportions and Manufacture of Geopolymer Pastes SSS-FA and Alkali-Activated Self-Compacting Concrete

For the study of the possible activation, different binder mixtures were formulated. A previous study evaluated the possibility of activation of the SSS through the use of NaOH 8 molar.

A liquid/solid ratio of 0.4 in all the binder mixtures was used, ensuring that similar, consistent values were used in all the mixtures. Two series were carried out with each of the processed SSS (70%SSS–30%FA and 35% SSS–65%FA), as well as two different curing conditions (Figure 5).

Once the pastes were manufactured, two systems of curing conditions were applied. First, the pastes were cured in an oven at 40 °C for 22 h, then they were de-moulded and cured in a wet chamber at 20 °C and 100% humidity. Like this first procedure, another curing system was carried out in which the initial temperature of the oven was 80 °C, the rest of the procedure was similar to the previous one.

The dosage percentages of each of the mixes and pastes manufactured and evaluated for subsequent selection are shown in Table 2, as well as the nomenclature applied.

After the analysis of the results of the mechanical behaviour of the manufactured alkaline-activated pastes shown in the section of test methods and results, it was decided to use three optimal mixtures (35SSS-NP, 35SSS-C, and 35SSS-B). Alkali-activated self-compacting concrete (SCC) was produced.

For the correct dosage of traditional SCC, it must be taken into account that the amount of cement, water, and fine aggregates must provide self-compacting characteristics of the SCC in relation to its flowability. Considering that the amount of water/cement must be controlled in order not to lose strength [42], the use of powdered pozzolanic fillers, fluidisers, and additives are used [43,44,45]. In this study, the cement and limestone filler used in the manufacture of traditional SCC is replaced by an alkali-activated binder with SSS-FA. Alkali-activated SCC was manufactured by replacing all limestone filler and 50% OPC with the binder SSS-FA. In addition, the amount of NaOH solution used to adjust the mixing water in each of the mixtures manufactured was considered, as well as the amount of admixture to achieve adequate flowability of SCC.

Three series of alkali-activated SCC were manufactured with the SSS-FA binder dosages that gave the best strength results. To evaluate the results obtained, a control mix was made.

Table 3 shows the dosages used in each of the mixtures for the manufacture of SCC.

The SCC manufacturing procedure was carried out in eight distinct phases.


Manufacture of binder using SSS-FA dissolution in NaOHAggregate-cement mixtureAdding binder to the mixAddition of remaining water + additiveConsistency measurement by flow extension testFilling of specimensCuring in an oven at 80 °C for 22 hDe-moulding and conservation in the curing chamber.


## 3. Test Methods and Results

In this work, an evaluation of physical, technical, and durability properties achieved by SCC manufactured with alkali-activated SSS-FA binder was carried out. Each of the tests carried out to evaluate the properties of SCC, as well as the applied standards and the dates of each test, are shown below (Table 4).

### 3.1. Flowability

The manufacture of SCC is associated with a detailed study of consistency [54]. Therefore, this property was analysed through three different methods: slump test according to UNE-EN 12350-10:2011, L-box according to UNE-EN 12350-10:2011, and J-ring according to UNE-EN 12350-12:2011 [48].

Table 5 shows the results obtained in terms of consistency for all the mixes made.

The results obtained showed that all the mixes made are within the slump limits established for SCC according to the three regulations applied.

Flow values for SCC manufactured with SSS-FA activated binder increased compared to those for the control SCC.

This increase in consistency with the addition of the activated binder can be attributed to the spherical shape of FA included in the binder, which leads to an increase in the volume of the paste and decreases friction in the blend [55,56,57].

L-Box tests were also performed for SCC mixtures to check their flowability. Table 5 shows that all mixtures have good capacity to pass through reinforcement bars without the need for compacting equipment, since all the values of H_2_/H_1_ blocking ratio are within the range of 0.88–0.93, within the limits established in the standards.

J-Ring tests were also performed. The values between the measurements obtained inside and outside the J-Ring ranged from 3.9 to 5.1 mm, being within the limits established in the regulations.

In relation to the influence of SSS treatment on the consistency of the activated SCC paste, it was observed that the application of crushing and burning increased in slump flow. All mixtures proved to be suitable for application on structural elements with reinforcement, according to the results obtained.

### 3.2. SSS-FA Binder—Compressive and Flexural Strength

In order to obtain SSS-FA mixes appropriate for use as cement substitutes in the manufacture of SCC, an initial study of the flexural and compressive strength of pastes made with different percentages of SS and FA activated with NaOH and cured under two different conditions was carried out.

Table 6 shows the results obtained for flexural and compressive strength measured at 7 and 28 days.

It can be observed that the flexural and compressive strength results increased under curing conditions at 80 °C (Table 6). Curing conditions at this temperature are more suitable for the activation of SSS. Similar results were obtained in previous studies [16], in which it was observed that pastes cured at 45 °C did not harden, while those cured at 80 °C could be removed from the moulds and showed resistance to mechanical tests.

The highest compressive strength results were obtained in pastes manufactured with 100% FA. As SSS was incorporated, the compressive strength decreased, reaching values of a reduction of approximately 55–75% in the pastes manufactured with 70% SSS and 30% FA.

Similar results were obtained in previous studies in which other types of metallurgical slag were used [22,58]. The authors showed that a higher substitution level, such as 60%, seemed to have a negative impact on the strength development in alkali-activated steel slag.

According to the result obtained, three mixes of alkali-activated SSS-FA were selected for the manufacture of SSC. Mixes that showed the best results in terms of compressive strength at 28 days were selected. The selected mixtures were 35SSS-NP, 35SSS-C, and 35SSS-B, underlined in Table 6.

### 3.3. SCC—Compressive Strength

The compressive strength was determined for three cubic samples of 100 mm for durations of 7 and 28 days according to UNE-EN 12390-3 [50].

According to the shown composition of the SSS (Table 1), the typology corresponds to a type of activation pattern (C-A-S-H), a material rich in calcium and silicon (SiO_2_ + CaO > 70%). It is activated under relatively moderate alkaline conditions [59,60].

Table 7 shows the results obtained for compressive strength of self-compacting concrete mixtures with activated slag. It was observed that a compressive strength between 34–35.5 MPa was obtained for mixtures with a cement content of 225 kg/m^3^ compared to the control mixture, which showed a compressive strength of 42.21 MPa for a cement content of 325 kg/m^3^.

All series manufactured resulted in a compressive strength in excess of 30 MPa at 28 days.

The use of activated SSS leads to a 50% reduction in cement. This reduction is associated with a loss of strength; however, this loss was 19.19% in 35SSS-NP, 16.94% in 35SSS-C, and 16.39% in 35SSS-B.

The reduction in compressive strength with the use of activated SSS as a cement substitute was less than that shown in other studies, in which the application of NaOH-activated blast furnace slag resulted in a loss of strength of approximately 65% over a control concrete [61].

The loss of relative strength in the application of SSS with respect to a control was much greater in the pastes evaluated than in the manufactured SCC (Figure 6).

The crushing and burning treatment of the SSS led to an improvement in the compressive strength. This improvement was more significant in the pastes than in the SCC, where the results were very close together in all three series.

### 3.4. SCC—Splitting Tensile Strength

A 28-day SCC tensile strength evaluation was conducted. This property does not depend exclusively on the strength of the aggregates used but also on the quality and type of connections established within them and the cementitious matrix, as well as micro-cracks or imperfections in the matrix [62].

The tensile strength of concrete is directly related to the behaviour of concrete against the initiation and propagation of cracks, steel reinforcement anchoring, or shearing. The tensile strength value can be predicted from the compressive strength of the concrete. The equations set out in Eurocode 2 [63] are used to predict the tensile strength. Both standards calculate the compressive strength according to Equation (1).
(1)fct,sp=13fcm−8MPa23

Figure 7 shows the results obtained for tensile strength, comparing them with the compressive strength of alkali-activated SCC. In addition, the theoretical tensile strength values calculated as a function of Equation (1) are shown.

In contrast to the results obtained for compressive strength, the tensile strength values were higher in all SCCs manufactured with binder SSS-FA compared to the control.

By applying the theoretical Equation (1), the results of control SCC could be predicted. However, it is not possible to apply it for alkali-activated SCC, as the use of SSS led to an increase in tensile strength unrelated to compressive strength.

Treatment of SSS by crushing and burning led to increased tensile strength values for SCC.

### 3.5. Water Absorption, Density, and Accessible Porosity

A study of the density, absorption, and porosity of SCC was carried out to see the influence of the application of alkali-activated SSS-FA binder to replace cement and limestone filler in the hardened state of the concrete.

To obtain these concrete properties, the apparent volume of the samples was first determined using a hydrostatic balance under saturation conditions. Then, they were placed in a vacuum chamber for 24 h to extract the occluded air. Over the next 24 h, the pores were filled with water by overpressure, and the specimens were kept submerged for a further 24 h to achieve full saturation of the specimens. The apparent volume and dry weight were obtained by this procedure when finally introduced into an oven at 110 °C until a constant weight was obtained. Once these data were obtained, the calculations established in the regulations were applied to obtain the accessible porosity, apparent density, and absorption coefficient.

The values obtained are shown in Table 8.

An increase in density was observed in SCC manufactured with SSS versus the control SCC. This increase was mainly due to the physical properties of SSS that show a higher density than cement and limestone filler (Table 1).

The absorption coefficient of alkali-activated SCC with SSS was substantially reduced compared to the results obtained with conventional SCC (control), as was observed with the open porosity values. Through the water absorption of the SCC, the degree of geopolymerisation and permeability of the material can be evaluated. A higher degree of geopolymerisation leads to a limited void fraction and an alkali-activated SCC internal matrix with low permeability [64].

Previous studies have shown that the use of fly ash in concrete reduced permeability parameters [65]. Similar results were shown in this study, in which the combined use of FA with SSS reduced porosity and absorption, a property that is directly related to good durability performance.

The application of activated SSS forms a gel that fills the pores and causes a denser microstructure that reduces the porosity of the concrete. Previous studies [66] concluded that the use of an alkaline solution providing more OH^−^ ions led to a reduction in permeability directly associated with improved mechanical properties. For this reason, it was observed that the microstructure formed in the alkali-activated SCC led to even better performance against tensile strength.

Crushing and burning treatment on alkaline-activated SSS reduced open porosity and SCC absorption. These values were reduced by approximately 50% when SSS-C were applied with respect to the control SCC. This is related to the pozzolanicity shown by SSS where a crushing treatment has been carried out (Figure 4). Positive pozzolanic activity led to a more stable microstructured geopolymerisation, which was reflected in the filling of pores in the internal matrix of the SCC and the reduction of porosity.

### 3.6. Macroporosity through Digital Image Analysis

Image analysis has been used in several works for the characterisation of concrete, for the analysis of its pore structure, and for the evaluation of permeability [67,68,69].

The samples for the image analysis were 50 mm sections in 100 × 100 mm cubic specimens. The surface was polished, and a white pigment was applied. The images were processed using image processing and analysis software (ImageJ™, Rasband W. A public domain Java image processing program. National Institute of Mental Health, Bethesda, MD, USA, 2008.) [70].

Figure 8 shows the processed digital images showing the pore distribution of control SCC and alkali-activated SCC. The self-compacting concrete made with SSS-C was analysed because it was the sample that showed the least porosity (35SSS-C).

Image analysis showed that the porosity of the SCC control was significantly higher than that of the SCC 35SSS-C manufactured with alkali-activated SSS binder. The pores in the 35SSS-NP samples were smaller in size and dispersed on the surface, while the pores in the control sample showed a larger size, with higher volume porosity and high connection between them.

SSS has a high Ca content, the slags are those that form part of the binder producing a dense and more compact matrix that reduces the number of pores mainly due to the fact that the gel formed in this case is C-A-S-H instead of N-A-S-H [71].

The reduction in open porosity in alkali-activated SCC specimens with SSS is directly associated with increased tensile strength values. A consolidation of the inner matrix produced greater cohesion in the specimens leading to increased strength (Figure 7).

### 3.7. Carbonation Depth

The carbonation study is necessary to evaluate the effects that an excess of humidity can cause in the steel reinforcement used in SCC. Initially, the SCC has a basic character medium (pH ≥ 12). Under these conditions, the steel reinforcement is passive and cannot be subject to corrosion. However, as time passes and as a result of the reaction of the portlandite with the ambient CO_2_, this pH drops to values below 9. In this situation, this concrete no longer protects the steel from corrosion, and if there is sufficient moisture, this will appear with the associated lesions.

The UNE 112011:2011 [53] method was used to determine the penetration of the carbonates. Prismatic samples of 100 × 100 mm, previously cured for 28 days in a wet chamber, were placed in an accelerated carbonation chamber with humidity conditions of 60%, temperature of 23 °C, and 5% of CO_2_.

Carbonation depth was evaluated for 1, 28, 56, and 90 days. An indicator applied to a specimen break with an atomiser (1% phenolphthalein solution in alcohol) was used to assess the change in pH.

The depth of carbonation was evaluated from the carbonation coefficient. This factor depends on the depth (Cd) compared to the square root of time in years (Figure 9).

All specimens manufactured with alkali-activated binder SSS-FA reduced the carbonation depth compared to the control. Less depth was observed when the SSS was subjected to crushing and burning treatments.

Figure 10 shows the 90-day carbonation depth in SCC control specimens (Figure 10a) versus the depth in 35SSS-C specimens (Figure 10b). Figure 10b shows that the use of alkali-activated SSS in the manufacture of SCC as a cement substitute reduced the carbonation depth of the specimen. SCC control showed an outer carbonation perimeter of approximately 15.5 mm; however, the carbonation depth in the 35SSS-C specimen was less than 3 mm.

The porosity of the material is an important factor that controls the depth of carbonation in SCC. As shown in Figure 8 and Table 8, the use of alkaline-activated SSS, in addition to reducing the cement content by 50% and the limestone filler content by 100%, leads to a reduction in the porosity of the material. Therefore, in these mixtures, the carbonation depth is lower, in contrast to the results shown in previous studies in which the use of FA and steel slag led to an increase in porosity and depth of carbonation in SCC [57,72]. This study shows that SSS in combination with FA reduces both properties.

In addition to the porosity of the material, the high Ca(OH)_2_ content has a buffering effect on the carbonation process by releasing hydroxide as the pH begins to decrease. A lower Ca(OH)_2_ content leads to a reduction in alkalinity and, therefore, to greater carbonation, worse mechanical behaviour, and the possibility of corrosion of the reinforcement [73,74].

This shows that the use of more alkali-activated binder in the mixtures leads to a reduction of the carbonation depth [75]. This is the result shown in Figure 9 and Figure 10, where alkaline-activated SCC specimens reduced carbonation penetration by 70–85% compared to that in control SCC.

## 4. Conclusions

This study has developed the possibility of applying alkali-activated SSS as a substitute for Portland cement and limestone filler in the manufacture of SCC. It has evaluated the influence of the treatment of SSS on the physical-chemical properties and the influence of their use on the mechanical properties and durability of alkali-activated SCC.

The following conclusions were obtained:i.SS presents a chemical composition suitable for the potential alkali activation. SSS are mainly composed of SiO_2_, CaO, and MgO. The application of crushing and burning treatments results in an increase in these three elements.

Although their mineralogy shows a crystalline structure that makes activation difficult, composed mainly of merwinite, calcium oxide, magnesium, silica, and akermanite (Ca-Mg-Si), pozzolanicity tests demonstrated the pozzolanic capacity when a crushing treatment (SSS-C) was carried out.


ii.Initial mechanical performance tests on pastes manufactured with SSS-FA and NaOH as an alkaline activator showed a reduction in strength as the percentage of SSS increased. It is important to point out that the mechanical behaviour improved with the appropriate curing conditions and the application of different treatments to the SSS.


Three optimal binder dosages were obtained for the manufacture of alkali-activated SCC (35SSS-NP, 35SSS-C, and 35SSS-B).


iii.The manufacture of SCC with the substitution of 50% cement by a binder manufactured with SSS and FA led to a loss of compressive strength of approximately 10–14%. The treatment of SSS improved the results of indirect traction and compression tests. All the series manufactured with SSS resulted in a compressive strength greater than 30 MPa at 28 days curing.iv.In terms of durability, SCC manufactured with alkali-activated SSS significantly reduced absorption and porosity parameters compared to a control SCC. Evaluation of macroporosity by image analysis showed a reduction in pore volume in alkali-activated SCC specimens with SSS compared to conventional SCC specimens. It also showed a lower pore connection.


This reduction in porosity and higher pH of alkali-activated SCC is directly related to the depth of carbonation. Alkali-activated SCC with SSS reduced up to 85% the carbonation depth. This factor, in the long term, leads to higher mechanical performance and less corrosion of steel reinforcements conventionally used in SCC.

In conclusion, the use of the alkali-activated SSS binder in combination with FA with NaOH as activator demonstrates the possibility of 50% substitution of Portland cement and limestone filler in the manufacture of SCC with particular curing conditions at 80 °C for the first 22 h.

The use of alkali-activated SSS reduces the compressive strength, although the values obtained exceed those established by regulations. Alkali-activated SCC in addition to reducing the cement content showed better tensile behaviour, lower porosity, and lower carbonation penetration than traditional SCC. This method is very effective in reducing CO_2_ emissions in Portland cement manufacturing and producing a stronger SCC in the long term.

Drying and grinding treatments of SSS (SSS-NP and SSS-C) are the most viable processes, reducing costs and pollution problems from burning and risks.

## Figures and Tables

**Figure 1 materials-14-03945-f001:**
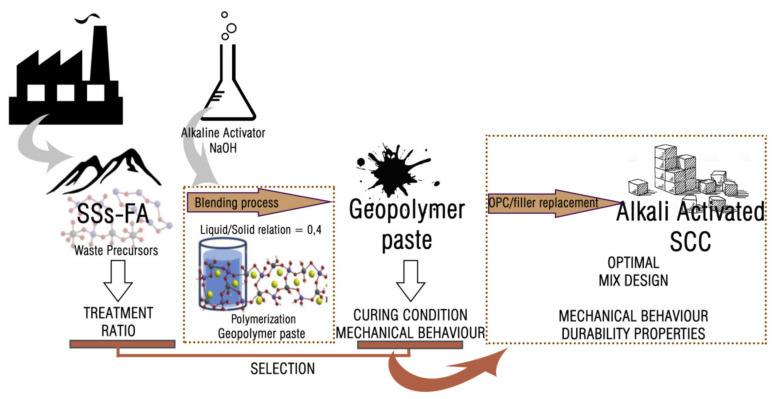
Diagram of the experimental methodology.

**Figure 2 materials-14-03945-f002:**
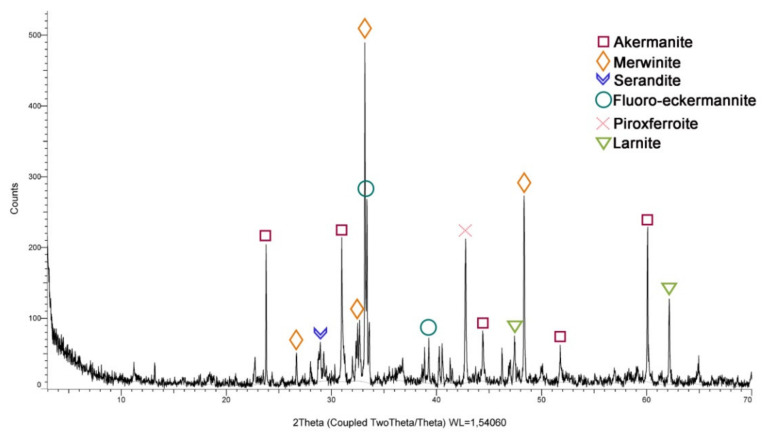
SSS diffractogram—XRD.

**Figure 3 materials-14-03945-f003:**
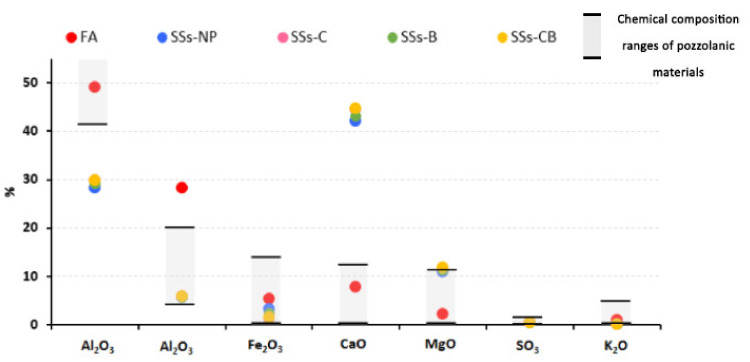
Chemical composition of FA and SSS in relation to ranges established for pozzolanic materials.

**Figure 4 materials-14-03945-f004:**
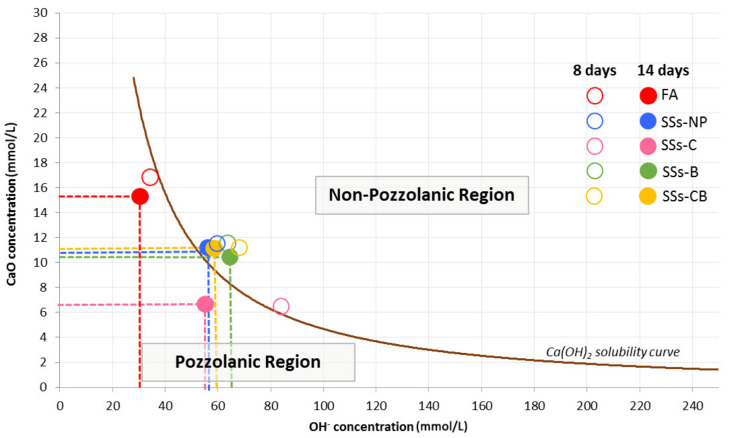
Result of Frattini test at 8 and 14 days of FA and SSS.

**Figure 5 materials-14-03945-f005:**
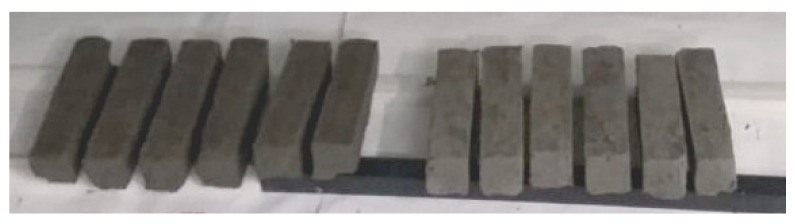
Alkali-activated SSS-FA pastes.

**Figure 6 materials-14-03945-f006:**
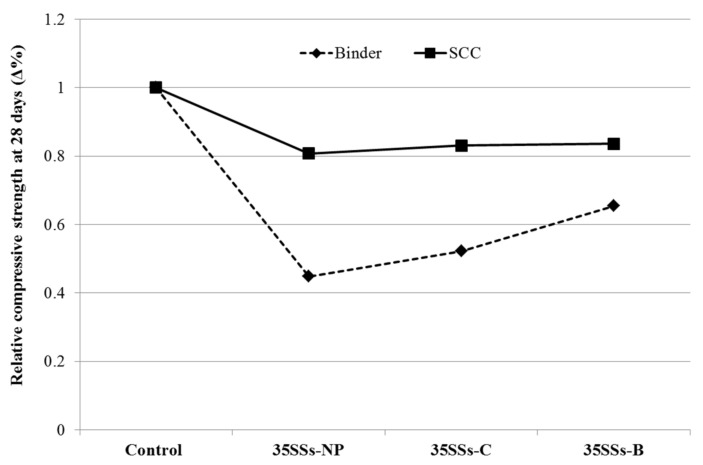
Relative compressive strength of binder and SCC at 28 days versus control.

**Figure 7 materials-14-03945-f007:**
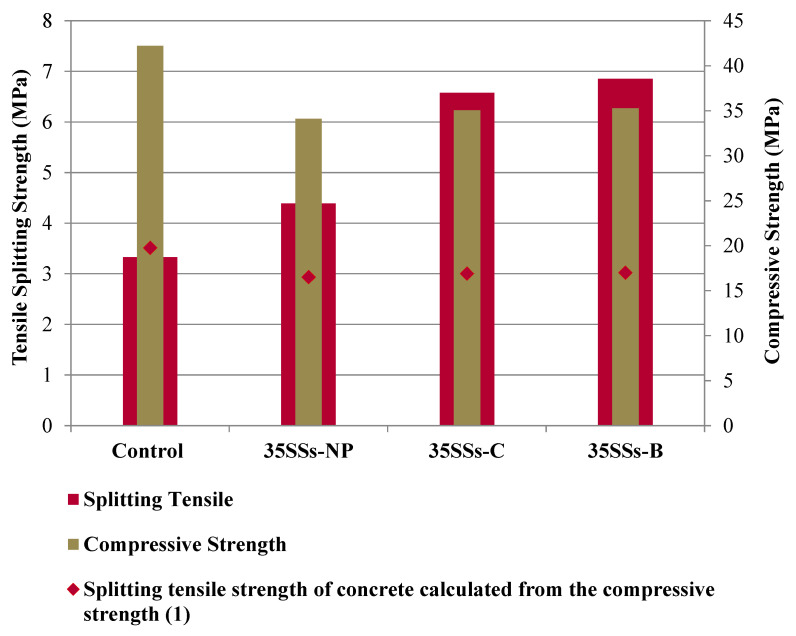
Tensile splitting strength versus compressive strength of SCC at 28 days.

**Figure 8 materials-14-03945-f008:**
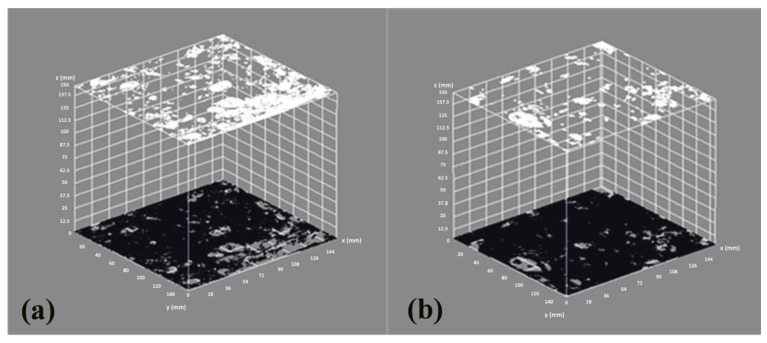
Pore structures of (**a**) SCC control and (**b**) 35SSS-C. Indicating in white the pores and in black the solid section.

**Figure 9 materials-14-03945-f009:**
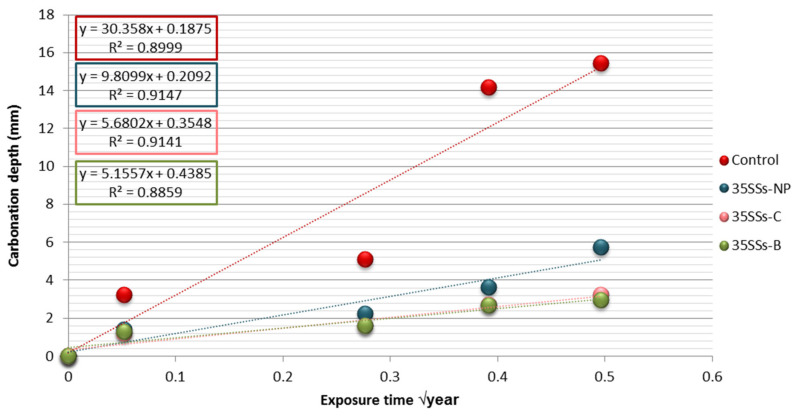
Carbonation depth according to √t of SCC control, 35SSS-NP, 35SSS-C, and 35SSS-B.

**Figure 10 materials-14-03945-f010:**
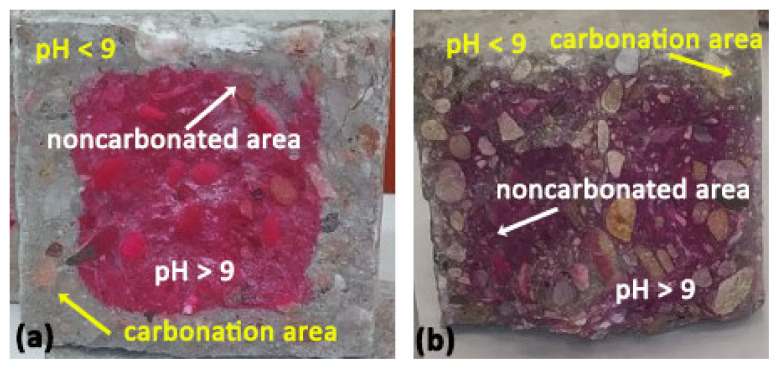
Transverse section of carbonated SCC specimens after 90 days of exposure: (**a**) SCC control and (**b**) 35SSS-C.

**Table 1 materials-14-03945-t001:** Physical and chemical properties of SSS.

		FA	SSS-NP	SSS-C	SSS-B	SSS-CB	Test method
Physical properties							
Density-SSD (kg/dm^3^)		1.28	2.06	1.8	1.69	1.66	UNE-EN 1097-6:2014 [32]
Water absorption (%)		4.35	6.12	5.31	5.44	5.41	
Chemical properties							
Elemental components (%)	SiO_2_	49.10	28.3	29.88	29.04	29.81	UNE-EN 196-2:2014 [33]
	Al_2_O_3_	28.33	5.64	5.73	5.64	5.83	
	Fe_2_O_3_	5.20	3.09	1.31	2.3	1.46	
	CaO	7.81	42.09	44.51	43.11	44.65	
	MgO	2.12	10.97	11.68	11.22	11.85	
	SO_3_	0.33	0.39	0.34	0.33	0.35	
	K_2_O	0.91	0	0	0	0	

**Table 2 materials-14-03945-t002:** Dosage of SSS-FA pastes.

SSS				FA	Nomenclature
SSS-NP	SSS-C	SSS-B	SSS-CB		
35	-	-	-	65	SSS-NP 35/65
-	35	-	-	65	SSS-C 35/65
-	-	35	-	65	SSS-B 35/65
-	-	-	35	65	SSS-CB 35/65
70	-	-	-	30	SSS-NP 70/30
-	70	-	-	30	SSS-C 70/30
-	-	70	-	30	SSS-B 70/30
-	-	-	70	30	SSS-CB 70/30

**Table 3 materials-14-03945-t003:** Mix proportion of alkali-activated SCC (kg/m^3^).

		CONTENT (kg/m^3^)								
		Natural Sand (0–40 mm)	Natural Aggregates (40–200 mm)	OPC	Filler	Admixture	Water	Binder 35SSS-FA		
								SSS-NP	SSS-C	SSS-B
SCC-30	Control	1000	700	325	125	3.41	195	-	-	-
	35SSS-NP	1000	700	227.5	-	1.65	144	227.5	-	-
	35SSS-C	1000	700	225	-	1.62	140	-	-	225
	35SSS-B	1000	700	225	-	1.62	140	-	225	-

**Table 4 materials-14-03945-t004:** Test methods.

		Test Method	Curing Time
Technological tests	Properties of fresh SCC		
	Flowability	UNE-EN 12350-8:2011 [46] UNE-EN 12350-10:2011 [47] UNE 12350-12:2011 [48]	0 days
	Properties of hardened SSS-FA binder		
	Compressive strength	UNE 196-1:2018 [49]	7, 28 days
	Flexural strength	UNE 196-1:2018 [49]	7, 28 days
	Properties of hardened SCC		
	Compressive strength	UNE 12390-3:2009 [50]	1, 7, 28 days
	Tensile splitting strength	UNE-EN 12390-6:2010 [51]	28 days
Durability properties	Water absorption, density, and accessible porosity	UNE 83980:2014 [52]	28 days
	Macroporosity through digital image analysis		28 days
	Depth carbonation	UNE 112011:2011 [53]	28, 56, 90 days

**Table 5 materials-14-03945-t005:** Fresh properties of SCC.

		Slump Test	L-Box Test			J-Ring
		D (mm)	T_200_ (s)	T_400_ (s)	H_2_/H_1_ (%)	DH = H_1_ − H_2_ (mm)
Compliance requirements		650–750	<1.5	<2.5	0.8–1	<10
(recommended)						
Control		712	1.23	1.97	0.88	5.1
35SSS-NP		715	1.36	2.12	0.93	3.9
35SSS-C		721	1.41	2.05	0.91	4.3
35SSS-B		719	1.48	2.23	0.91	4.5

**Table 6 materials-14-03945-t006:** Flexural and compressive strength of alkali-activated slag pastes.

	CURING CONDITION 40 °C (22 Initial Hours)				CURING CONDITION 80 °C (22 Initial Hours)			
	Flexural Strength (MPa)		Compressive Strength (MPa)		Flexural Strength (MPa)		Compressive Strength (MPa)	
	7D	28D	7D	28D	7D	28D	7D	28D
100FA-CONTROL	1.51	5.01	3.33	11.91	10.86	4.36	32.79	44.67
35SSS-NP	0.31	2.15	2.31	4.42	4.35	6.44	13.50	20.04
70SSS-NP	1.3	1.33	0.64	1.03	3.68	4.35	7.42	17.49
35SSS-C	0.92	2.34	0.34	3.01	0.96	3.63	6.48	23.32
70SSS-C	0.66	1.32	1.41	1.72	1.20	2.80	5.96	19.55
35SSS-B	0.93	1.98	1.56	1.79	2.48	4.39	19.59	29.26
70SSS-B	1.23	1.57	0.83	1.06	0.46	2.35	5.54	11.28
35SSS-CB	0.37	0.73	0.61	2.52	2.15	5.01	12.15	18.75
70SSS-CB	0.187	1.85	1.06	3.43	1.06	3.75	9.63	15.25

**Table 7 materials-14-03945-t007:** Compressive strength of alkali-activated SCC-30.

	Compressive Strength (MPa)	
	7D	28D
Control	25.86	42.21
35SSS-NP	17.44	34.11
35SSS-C	21.2	35.06
35SSS-B	22.77	35.29

**Table 8 materials-14-03945-t008:** Density, absorption, and porosity of alkali-activated SCC.

	Control	35SSS-NP	35SSS-C	35SSS-B
Bulk density (kg/dm^3^)	2.28	2.36	2.31	2.3
Absorption coefficient (%wt.)	9.11	6.29	4.48	5.79
Open porosity (%vol.)	20.78	14.86	10.39	13.33

## Data Availability

Not applicable.

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
