# Peer review of "Alkali-Activated Stainless Steel Slag as a Cementitious Material in the Manufacture of Self-Compacting Concrete"

_materials, 2021, doi:10.3390/ma14143945_

Round 1

Reviewer 1 Report

A nice and well-presented work. The reviewer has a few comments/suggestions on this manuscript.

1) Page 3, line 120-122, that should be one sentence, not two.

2) Still page 3, Table 1, why the density has a huge difference?

3) Page 5, line 157-158, why the author says so? Please add more explanations to it. 

4) Please also carefully check the manuscript, there are still many small mistakes in it. 

Author Response

A nice and well-presented work. The reviewer has a few comments/suggestions on this manuscript.

  • Page 3, line 120-122, that should be one sentence, not two.

Thank you for your comment. It has been corrected.

  • Still page 3, Table 1, why the density has a huge difference?

 The crushing process and especially the subsequent screening of the SSs involves the removal of high density metals that could not be crushed by the mechanical grinding measures used. The same applies to the combustion process. The temperatures reached in the furnace cause the decomposition of the metal compounds present in the slag, resulting in a decrease in density.

  • Page 5, line 157-158, why the author says so? Please add more explanations to it. 

Thank you for your comment. It has been explained in the text.

The explanation made was not correct. The corrected explanation is shown below.

“According to Frattini's test it can be seen from figure 1 that the data obtained on the variation of OH- and CaO show that the pozzolanic activity of SSs-C is high, while SSs-NP, SSs-B, SSs-CB and FA are less reactive. SSs-C showed low OH- and CaO values that place this slag in the "pozzolanic region" established by Frattini (Fig.4). Higher values of OH- and CaO concentrations in the rest of the slags analysed place them in the "non-pozzolanic region" showing a lower activation potential”.

  • Please also carefully check the manuscript, there are still many small mistakes in it.

 Thank you, the manuscript has been revised and several mistakes have been corrected.

Reviewer 2 Report

In the research paper „Alkali-activated stainless steel slag as a cementitious material in the manufacture of self-compacting concrete” the Authors used an activated stainless steel slag as a substitute for cement and limestone filler. The paper is written in clear and mostly understandable language. The experiments were prepared well and the conclusions follow obtained results. However, some work must be performed before this manuscript can be accepted.

Below are my detailed comments:

Line 31 „... due to lower consumption”.  Emission?

Line 54-55 „Using this” as?

The Paragraph in lines 54-65 should be rephrased to be more clear and informative. E.g. „compressive strength” of geopolymers/concrete?

Line 115: Why the SSs treatment decreased the density and water absorption?

Tab.1 Water Absorption (%)?

Lines 126-127: How it is that possible to transform the crystalline slag into amorphous by combining slag with FA? The Authors should give a detailed explanation of this process.

Line 157: Symbols should be within brackets.

Line 172: What was the reasoning behind choosing such SSs and FA percentages?

Tab. 3. The size of the Natural Aggregates should be provided.

Lines 222-223: „slump test, L-box and J-ring”. Either a short description or a reference to the description of those methods should be provided.

Line 231: A spherical shape of SSs…, the meaning of this sentence is unclear. The Authors stated earlier that SSs had a crystalline structure. Spherical shape crystals?

Line 261-263: If it is already known from the literature [22,48] that „a higher substitution level, such as 60%, seems to be negative to the strength development…”. Then what was the point of testing the 70% substitution in this research? Especially when results in Tab.6 only confirms what is already known.

Line 245: Optimization  with only two values of 35 and 70% can hardly be called optimization. Would not it be better to pick a few values below the 60% limit, and perform the proper optimization tests? Results in Tab. 6 shows that a 100FA-CONTROL have usually over 2 times higher strength than that with 35% and even more with 70%. Based on these results it looks like the most optimal case was 0% SSs and 100%FA. If so, what is the point of adding SSs if it has such a negative impact?

Line 266: Why the selection criteria were set as 20 MPa and not e.g. 19.5 MPa or 20.1 MPa?

Tab. 6, In the first column the underlining is missing.

Tab. 7, MPa might be moved to a headline, to avoid repeating it for all values.

Fig. 6. Vertical axis label: the word „relative” is missing.

In my opinion, the paragraph in lines 325 to 333 is more suitable to chapter 2, than here.

Units are not consistent throughout the manuscript. In Tab.1 the Authors provide density as kg/dm3, e.g. Tab. 3 as kg/m3 and Tab. 8 as g/cm3.

Fig. 10. The pictures look unnatural, how were they obtained ? Was numerical postprocessing involved ? if so, what was the exact procedure? Also, the signs ‘>’ and ‘<’ should be reverted i.e. in the non-carbonated zone it should be pH > 9 and in the carbonated pH < 9).

Line 415: Not only the porosity of the material controls the depth in this case. NaOH was added with SCC and so the pH was higher at the beginning and that affect the carbonation level as well.

Lines 438-445: This is not a conclusion based on performed/presented research in this manuscript.

Line 464: The sentence: „This reduction in porosity is directly related to the depth of carbonation.” Should be replaced with: „This reduction in porosity and higher pH of alkali-activated SCC is directly related to the depth of carbonation.

Author Response

In the research paper „Alkali-activated stainless steel slag as a cementitious material in the manufacture of self-compacting concrete” the Authors used an activated stainless steel slag as a substitute for cement and limestone filler. The paper is written in clear and mostly understandable language. The experiments were prepared well and the conclusions follow obtained results. However, some work must be performed before this manuscript can be accepted.
Below are my detailed comments:
1)    Line 31 „... due to lower consumption”.  Emission?
Thank you for your comment. The text has been corrected

2)    Line 54-55 „Using this” as?
Thank you, this sentence has been rewritten in the text to make it easier to understand.

3)    The Paragraph in lines 54-65 should be rephrased to be more clear and informative. E.g. „compressive strength” of geopolymers/concrete?
Thank you for your comment. The paragraph has been modified so that the reader can understand it.
New clarifications have been included in the manuscript and some sentences have been rewritten for better understanding.  

4)    Line 115: Why the SSs treatment decreased the density and water absorption?

The crushing process and especially the subsequent screening of the SSs involves the removal of high density metals that could not be crushed by the mechanical grinding measures used. The same applies to the combustion process. The temperatures reached in the furnace cause the decomposition of the metal compounds present in the slag, resulting in a decrease in density. The same occurs with the absorption of water. The presence of coarse particles and aggregates combined by different steel slags leads to a higher absorption. When a crushing or burning process and subsequent screening is carried out, these particles are disintegrated, thus reducing the water absorption obtained by the unprocessed material.

5)    Tab.1 Water Absorption (%)?

Thank you. The text in Table 1 has been corrected.

6)    Lines 126-127: How it is that possible to transform the crystalline slag into amorphous by combining slag with FA? The Authors should give a detailed explanation of this process.
It is not really that the SS is transformed into an amorphous structure. It is that the combination of SSs with FA generates a final product that on XRD analysis is characterised as amorphous.
This has been clarified in the text.
7)    Line 157: Symbols should be within brackets.
Thank you, the symbols have been corrected.

8)    Line 172: What was the reasoning behind choosing such SSs and FA percentages?
The idea of this work is to use the maximum possible amount of SSs in order to obtain adequate mechanical and durability results in the manufacture of self-compacting concrete. 
It has been shown that the use of FA in cement mixtures results in a material with high compressive strength. In this work, the aim is to reduce the use of FA by using SSs. 
Before choosing these percentages, a laboratory study was carried out with higher combinations of SSs, obtaining very low compressive strength results. Therefore, the dosages were adjusted until the optimum combination was found that combined a higher use of SSs with adequate mechanical strength.

9)    Tab. 3. The size of the Natural Aggregates should be provided.
Thank you for your comment. The particle size of the natural aggregates has been included in Table 3.

10)    Lines 222-223: „slump test, L-box and J-ring”. Either a short description or a reference to the description of those methods should be provided.
The UNE standard applied to carry out each of the tests has been included in the text.

11)    Line 231: A spherical shape of SSs…, the meaning of this sentence is unclear. The Authors stated earlier that SSs had a crystalline structure. Spherical shape crystals?
The authors want to refer to the spherical shape of the FA. The text has been corrected to make sense of this sentence.

12)    Line 261-263: If it is already known from the literature [22,48] that „a higher substitution level, such as 60%, seems to be negative to the strength development…”. Then what was the point of testing the 70% substitution in this research? Especially when results in Tab.6 only confirms what is already known.

The study has been carried out with a higher percentage because in previous studies the slags analysed were different to the type of SSs. This study is characterised for being innovative in the use of stainless steel slags. As there is not much previous bibliography, it is necessary to compare it with studies of other types of metallurgical slags and to carry out new mixtures and methodologies to prove its potential.
This has been clarified in the text.

13)    Line 245: Optimization  with only two values of 35 and 70% can hardly be called optimization. Would not it be better to pick a few values below the 60% limit, and perform the proper optimization tests? Results in Tab. 6 shows that a 100FA-CONTROL have usually over 2 times higher strength than that with 35% and even more with 70%. Based on these results it looks like the most optimal case was 0% SSs and 100%FA. If so, what is the point of adding SSs if it has such a negative impact?

As mentioned previously, this study looks at the use of SSs in the manufacture of self-compacting concrete. 
It is correct that the use of FA results in higher compressive strength, but the application of FA is not a scientific novelty. FA are ashes that have been applied for years, so the mixture 0% SSs and 100%FA can be considered as a control. 
Actually, the reviewer is right that the aim is not to optimise the mixes, because the results of many different mixes are not shown, but the aim is to find the best results obtained among the binders produced. These binders will be used later in the production of SCC as a cement substitute.
This aspect has been clarified in the text.

14)    Line 266: Why the selection criteria were set as 20 MPa and not e.g. 19.5 MPa or 20.1 MPa?

For the selection of the mixes, the 3 binders with the best compressive strength were used as criteria. In the previous text this sample selection was expressed in the wrong way. 
This has been corrected in the manuscript.

15)    Tab. 6, In the first column the underlining is missing.

Thank you for your comment.
Only compressive strength values exceeding the established criteria have been underlined.
In this case, they are only obtained for some binders cured at 80°C (last column).
Underlined values that were not correct have been corrected in the text.

16)    Tab. 7, MPa might be moved to a headline, to avoid repeating it for all values.
Thank you for your comment. The table has been modified    

17)    Fig. 6. Vertical axis label: the word „relative” is missing.

Thank you, the text on the axis of the figure has been corrected.

18)    In my opinion, the paragraph in lines 325 to 333 is more suitable to chapter 2, than here.

Thank you for your comment. We consider it appropriate to leave the paragraph in the current section because it refers to the properties of concrete. Section 2 discusses material properties, therefore the authors do not consider it appropriate to combine material properties with concrete properties.

19)    Units are not consistent throughout the manuscript. In Tab.1 the Authors provide density as kg/dm3, e.g. Tab. 3 as kg/m3 and Tab. 8 as g/cm3.

Thank you for your comment. 
The units in Table 1 and 8 have been unified as they refer to the same properties. The unit used has been kg/dm3, which is the unit commonly used to express this property.
However, in Table 3, the unit kg/m3 has been retained, as this is the usual way of expressing the dosages in concrete manufacture.

20)    Fig. 10. The pictures look unnatural, how were they obtained ? Was numerical postprocessing involved ? if so, what was the exact procedure? Also, the signs ‘>’ and ‘<’ should be reverted i.e. in the non-carbonated zone it should be pH > 9 and in the carbonated pH < 9).

Thank you for indicating the error in the <,> sign. This error has been corrected.
Regarding the image, it has been treated with an image editing program to achieve a higher contrast between the areas.
The unprocessed image is shown below. If you consider it convenient, it will be replaced.

21)    Line 415: Not only the porosity of the material controls the depth in this case. NaOH was added with SCC and so the pH was higher at the beginning and that affect the carbonation level as well.

You are right that the addition of NaOH to produce the activation of SSs increases the pH and may affect carbonation. However, the control SCC ( manufactured without NaOH) showed higher pH values and therefore higher carbonation than SCC manufactured with alkaline activated SSs (with NaOH). Therefore, the presence of NaOH does not directly affect the carbonation of SCC in the short and medium term.

22)    Lines 438-445: This is not a conclusion based on performed/presented research in this manuscript.

Thank you for your feedback.
The authors think it is important to show as a conclusion that the application of a treatment to the SSs implies a modification of their main components. As well as, that treating SSs increases the pozzolality of the material, according to the data shown in section 2.1.

23)    Line 464: The sentence: „This reduction in porosity is directly related to the depth of carbonation.” Should be replaced with: „This reduction in porosity and higher pH of alkali-activated SCC is directly related to the depth of carbonation.

Thank you. The sentence has been modified in the manuscript

Reviewer 3 Report

This study treats an interesting topic and it was conducted scientifically and systematically. Sufficient references were given in the explanations of the obtained results. Analyses were carried out in depth. I have a favorable stance for the publication of the manuscript after completing recommended corrections below.

  1. Sentences in line 56-57 should be merged. Currently the flow is problematic.
  2. Line 121-122 typing mistake.
  3. Caption of figure 2 can be centered and DRX can be changed to XRD (optional).
  4. Table 4, first column “durability” line should be corrected.
  5. Line 255 “. (dot)” is missing at the end of the sentence.
  6. In subsection 3.3, a curve for the compressive behavior can be given for a single specimen for each group to show the mechanical behavior more explicitly.
  7. Number given in Figure 8 cannot be clearly seen. Another image with a higher resolution can be given. In addition, gray background can be removed (optional).

Author Response

This study treats an interesting topic and it was conducted scientifically and systematically. Sufficient references were given in the explanations of the obtained results. Analyses were carried out in depth. I have a favorable stance for the publication of the manuscript after completing recommended corrections below.
1)     Sentences in line 56-57 should be merged. Currently the flow is problematic.

Thank you for your comment. The text has been modified in the manuscript.

2)    Line 121-122 typing mistake.

Thank you. It has been corrected.

3)    Caption of figure 2 can be centered and DRX can be changed to XRD (optional).

Thank you. The caption has been modified.

4)    Table 4, first column “durability” line should be corrected.

Thank you. There has been an error in the PDF printout.

5)    Line 255 “. (dot)” is missing at the end of the sentence.

Thank you, the text has been corrected.

6)    In subsection 3.3, a curve for the compressive behavior can be given for a single specimen for each group to show the mechanical behavior more explicitly.

Thank you for your comment. Below is a figure showing the progression curve of the compressive strength values obtained.
The authors consider that it is duplicative to include the table with data and the curve, as it does not provide additional information.
If the reviewer considers it necessary to introduce such a figure, the manuscript will be modified.

7)    Number given in Figure 8 cannot be clearly seen. Another image with a higher resolution can be given. In addition, gray background can be removed (optional).
Thank you. The figure has been improved and modified in the manuscript.

Round 2

Reviewer 2 Report

line 61 -  SiO2 -> SiO2
line 130-132 -  so the combination of SSs with FA generates the final product which is a mixture of crystalline and amorphous which XRD analysis characterise as amorphous ?

line 135 -  (Ca3Mg (SiO4)2) -> the space is obsolete

Figure 10 - Maybe I just got used to unprocessed images, so the edited looks a bit unnatural, therefore I would choose the original ones. Nevertheless, this image still needs some improvement. "Passive area" should be replaced with "noncarbonated area" ("passive area" term is more frequently used with regards to the state of the reinforcement steel) and also "carbonation front" with "carbonation area".

Author Response

Dear Ms. Hristina Stankovic:

Attached please find our revised manuscript materials-1275551 entitled:

“ALKALI-ACTIVATED STAINLESS STEEL SLAG AS A CEMENTITIOUS MATERIAL IN
THE MANUFACTURE OF SELF-COMPACTING CONCRETE”.

We are grateful for the constructive comments on this second revision, which we believe have allowed us to substantially improve our manuscript.

We have thoroughly revised the manuscript. I would like to first address the principle concerns (in blue) and indicate how we have dealt with them. Secondly, changes made to the manuscript are indicated in red.  The answers to the various suggestions/comments from the reviewer are given below:

Reviewers' comments:

  • Line 61 -  SiO2 -> SiO2

Thank you. It has been corrected.

  • Line 130-132 -  so the combination of SSs with FA generates the final product which is a mixture of crystalline and amorphous which XRD analysis characterise as amorphous ?

The authors have decided to delete this statement from the manuscript. Because no scientific evidence is provided to corroborate the transformation from crystalline to amorphous phase when combining SSs with FA.

We apologise for the error of introducing a statement that has not been verified by laboratory tests.

  • Line 135 -  (Ca3Mg (SiO4)2) -> the space is obsolete

Thank you. It has been corrected.

  • Figure 10 - Maybe I just got used to unprocessed images, so the edited looks a bit unnatural, therefore I would choose the original ones. Nevertheless, this image still needs some improvement. "Passive area" should be replaced with "non-carbonated area" ("passive area" term is more frequently used with regards to the state of the reinforcement steel) and also "carbonation front" with "carbonation area".

 Thank you for your comment. We have modified the text in the image and included the unprocessed image in the manuscript.

After complete the revision process, we hope that the revised manuscript does now fully meet the criteria and conditions for publication in Materials. Thank you very much for your efforts concerning our manuscript. 

Yours sincerely,

Ph D. Francisco Agrela

Construction Engineering. University of Cordoba.